# Fast and strong amplifiers of natural selection

Josef Tkadlec [1,4✉], Andreas Pavlogiannis[2,4], Krishnendu Chatterjee[3] & Martin A. Nowak [1]

Selection and random drift determine the probability that novel mutations fixate in a population. Population structure is known to affect the dynamics of the evolutionary process. Amplifiers of selection are population structures that increase the fixation probability of beneficial mutants compared to well-mixed populations. Over the past 15 years, extensive research has produced remarkable structures called strong amplifiers which guarantee that every beneficial mutation fixates with high probability. But strong amplification has come at the cost of considerably delaying the fixation event, which can slow down the overall rate of evolution. However, the precise relationship between fixation probability and time has remained elusive. Here we characterize the slowdown effect of strong amplification. First, we prove that all strong amplifiers must delay the fixation event at least to some extent. Second, we construct strong amplifiers that delay the fixation event only marginally as compared to the well-mixed populations. Our results thus establish a tight relationship between fixation probability and time: Strong amplification always comes at a cost of a slowdown, but more than a marginal slowdown is not needed.

[1] Department of Mathematics, Harvard University, Cambridge, MA 02138, USA. [2] Department of Computer Science, Aarhus University, Aabogade 34, 8200 Aarhus, Denmark. [3] Institute of Science and Technology Austria, Am Campus 1, 3400 Klosterneuburg, Austria. [4]These authors contributed equally: Josef Tkadlec, Andreas Pavlogiannis. ✉email: josef.tkadlec@gmail.com

Populations evolve by accumulating novel mutations. Once a new mutant arises, selection and random drift govern the dynamics of spread until the mutant either fixates or becomes extinct. Natural selection favors the proliferation of advantageous mutants. But their fixation is not guaranteed, because randomness could wipe out the mutant clone before it reaches a sizable part of the population. A key quantity in this process is the fixation probability, that is, the probability that the lineage of a new advantageous mutant eventually takes over the whole population[1–3]. The fixation probability is a central concept in evolutionary theory and it is often used to characterize the speed of evolution, as high fixation probability accelerates the evolutionary process[4–6].

Fixation probabilities are often calculated for the idealized case of well-mixed populations[7–9]. In a well-mixed population, all individuals are in equivalent positions and compete uniformly with all other individuals. Biological populations are often not well-mixed, but rather follow some spatial (or other) arrangements. Population structure can affect evolutionary dynamics, as invading mutants arise in specific locations and spread locally[10–13]. Although—at first sight—this local spread might appear to impede the fixation of new mutants, certain population structures are known to increase the fixation probability of advantageous mutants compared to a well-mixed population[14–19]. This remarkable property of amplifying the selective advantage of mutants has been a subject of extensive studies, and several amplifying structures have been discovered over the years[20–23]. The epitome of this effect is achieved by strong amplifiers, which are population structures that, in the limit of large population size, guarantee fixation of advantageous mutants with high probability, regardless of how small the mutant's fitness advantage is over the resident[24–29].

Besides fixation probability, another important quantity in the evolutionary process is the 'fixation time', that is, the expected number of generations from the arrival of the mutant until the population resolves to a homogeneous state (which is the fixation of either the mutant or the residents)[30–32].

Amplifiers are known to substantially increase the fixation time as compared to well-mixed populations[33–36], which in turn limits their role as accelerators of evolution due to the following effect: The evolutionary process operates in two very distinct regimes depending on the magnitude of the mutation rate, relative to the fixation time. When the mutation rate is relatively high, multiple independent mutations typically compete for fixation, known as clonal interference[37–42]. In this regime, the rate of evolution is primarily determined by the fixation time (as opposed to the fixation probability). In contrast, when the mutation rate is relatively low, mutations occur mostly sequentially. The rate of evolution is now primarily determined by the probability that a new mutation fixates before the next one occurs, and thus amplifiers accelerate evolution. Since the threshold between these two regimes is determined by the fixation time, amplifiers with long fixation times realize their effect only in very low mutation rates. On the other hand, amplification combined with short fixation time acts in a broader range of mutation rates, but thus far has remained elusive.

A standard model of stochastic evolutionary dynamics is the Moran process[7]. The fitness advantage of a mutant invading a resident population is specified by a parameter $r > 1$, while residents have fitness normalized to 1. For a well-mixed population of size $N$, an invading mutant fixates with probability $(1 − 1/r)/(1 − 1/r^N)$. For large $N$, the fixation probability converges to $1 − 1/r$ and the process runs for approximately $(1 + 1/r) \cdot \log N$ generations. Ignoring the constant factor $1 + 1/r$ and focusing on the dependence on the population size $N$, we say that the 'timescale of fixation' on the complete graph is (of the order of) $\log N$ generations.

The Moran process can be adapted to structured populations[24,43]. Population structure is represented by a graph,

with each node occupied by a single individual and edges marking where a reproducing individual can place its offspring[44]. The well-mixed population corresponds to the complete graph $K_N$. The prime example of an amplifier is the Star graph $S_N$, which increases the fixation probability to approximately $1 − 1/r^2$, while the timescale is increased to $N \cdot \log N$ generations[14,21,22,36] from the $\log N$ generations of $K_N$. Instances of strong amplifiers, such as superstars, funnels, and megastars have also been discovered, with the remarkable property that for any $r > 1$ the fixation probability approaches 1 in the limit of large $N$, hence fixation is reached almost surely[24–26]. However, these structures operate on long timescales where mutants and residents coexist for many generations until the population reaches a homogeneous state. The search for faster strong amplifiers has lead to structures, such as the Dense Incubators $D_N$[27,29], with timescale that is polynomial in $N$[45,46]. Since well-mixed populations resolve in $\log N$ generations, strong amplification still comes at the cost of a substantial increase in timescale. Figure 1 illustrates the landscape of the probability-timescale trade-off for some common structures.

All existing trade-offs between probability and time highlight two fundamental questions about the evolutionary process: (1) are there strong amplifiers with timescale as short as that of well-mixed populations? And, if not, (2) what is the smallest possible slowdown for which strong amplification is possible?

Here we answer those questions by establishing a tight relationship between the probability of fixation and the timescale of fixation. First, we show that the timescale of any strong amplifier is larger than the timescale of the well-mixed population. Second, we construct strong amplifiers whose timescale is only marginally longer than that of the well-mixed population. Therefore, we show that strong amplification always comes at the cost of a slowdown, but more than a marginal slowdown is not needed.

## Results

**Model.** In the framework of evolutionary graph theory, a population consisting of $N$ individuals (each either a 'mutant' or a 'resident') is spread over $N$ nodes of a fixed connected graph $G_N$.

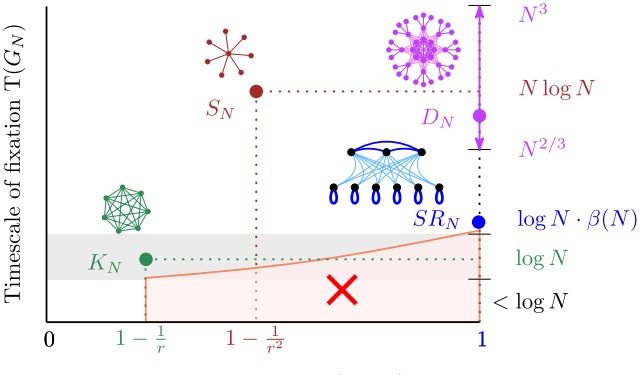

**Fig. 1 Probability-timescale landscape of population structures.** On a large well-mixed population ($K_N$, green), a single mutant with a fixed relative fitness advantage $r > 1$ fixates with a constant probability $1 − 1/r$ after a number of generations that scales logarithmically with the population size $N$. On a Star graph ($S_N$, brown), the fixation probability increases to $1 − 1/r^2$, but the process takes roughly $N\log N$ generations, an exponential slowdown in timescale relative to $K_N$. Dense Incubators ($D_N$, purple) push the fixation probability to 1 within a polynomial timescale. Selection reactors ($SR_N$, blue), introduced here (see Theorem 2), guarantee fixation with high probability while almost matching the timescale of well-mixed populations (here $\beta(N)$ is an arbitrarily slowly growing unbounded function). No population structure can appear in the red region (see Theorem 1).

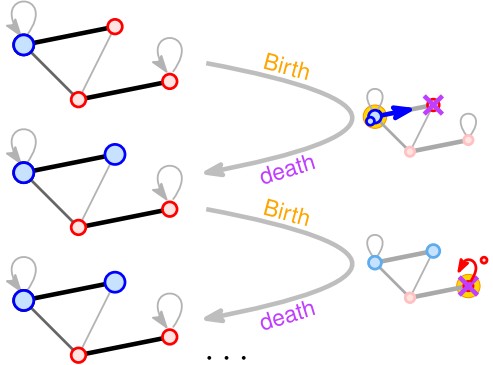

**Fig. 2 Moran birth–death process on a population structure.** Residents (red, with fitness 1) and mutants (blue, with fitness advantage $r > 1$) are spread over nodes of a graph. Its nodes represent sites and its weighted edges represent migration rates. At each discrete time step, one individual reproduces ("Birth", yellow) and places its offspring on a neighboring node, replacing its initial occupant ("Death", purple). Eventually, either the mutants or the residents fixate on the whole population.

The nodes of $G_N$ represent the sites and the edges (possibly directed, weighted, and including self-loops) represent the migration rates between neighboring sites. Each mutant has fixed fitness $r > 1$ and each resident has fitness 1. The individuals reproduce asexually with a rate proportional to their fitness, and the produced offspring migrates along an edge, replacing the original inhabitant of that site (see Fig. 2).

Formally, the birth–death Moran process proceeds in discrete steps according to the following stochastic rule:

(1) (Birth) Select an individual for reproduction, with probability proportional to its fitness. That individual, say at node $u$, produces an offspring that is a copy of itself.

(2) (Death) Select a neighboring node $v$ of $u$, with probability proportional to the edge weight $w_{u,v}$. The offspring created at node $u$ migrates to node $v$ and replaces its initial inhabitant.

The process ends once all individuals are of the same type—either all mutants ('fixation' occurs) or all residents ('extinction' occurs). The 'fixation probability' $fp(G_N, r)$ is the probability that a single mutant with fitness $r > 1$ appearing at a node of $G_N$ selected uniformly at random eventually fixates. Similarly, we denote by $ep(G_N, r) = 1 - fp(G_N, r)$ the 'extinction probability' and by $FT(G_N, r)$ the 'fixation time', that is, the (expected) number of generations until the process ends (here each generation consists of $N$ individual steps). Finally, ignoring the lower-order terms in $FT(G_N, r)$ and a multiplicative constant that possibly depends on $r$ but not on $N$, we obtain the 'timescale of fixation' $T(G_N)$. See Supplementary Note 1 for further discussion.

**Amplifiers and strong amplifiers.** The well-mixed population is modeled by the complete graph $K_N$ with all edges of unit weight. It is known[2,7,36] that for any fixed $r > 1$ we have

$$fp(K_N, r) \to 1 - 1/r \quad \text{and} \quad T(K_N) = \log N \qquad (1)$$

as $N \to \infty$.

This gives a natural baseline to which one can compare other population structures. In particular, a population structure $G_N$ with $N$ nodes is called an 'amplifier' if it increases the fixation probability of advantageous mutations as compared to the complete graph $G_N$ of the same size, that is, if $fp(G_N, r) > fp(K_N, r)$ for all $r > 1$. Even more strongly, a sequence $\{G_N\}_{N=1}^{\infty}$ of population structures of increasing size is called a 'strong amplifier' (also known as a 'superamplifier') if, for any $r > 1$, it satisfies $fp(G_N, r) \to_{N \to \infty} 1$.

Strong amplification is, in a sense, the strongest possible form of amplification as it ensures the fixation of advantageous mutants with probability close to 1.

A prominent example of an amplifier is the Star graph $S_N$, consisting of one center node and $N - 1$ leaf nodes connected to the center. It is known[21,33,36,47] that for any fixed $r > 1$ the Star graph satisfies

$$fp(S_N, r) \to 1 - 1/r^2 \quad \text{and} \quad T(S_N) = N \log N \qquad (2)$$

as $N \to \infty$. For example, when $r = 1 + s$, where $s$ is small, large Stars roughly double the fixation probability at the cost of lengthening the timescale from logarithmic to polynomial. A similar increase to both fixation probability and timescale has been known for bipartite graphs, and extensive numerical studies on small graphs show a strong positive correlation between fixation probability and the timescale[36,48].

Strongly amplifying structures have also been discovered over the years, such as superstars, funnels, and megastars[24,25], while it has been recently shown that strong amplification can be obtained out of almost every structure if one adjusts the migration rates between neighboring sites[28]. All these results present families $\{G_N\}_{N=1}^{\infty}$ with the property that $fp(G_N, r) \to_{N \to \infty} 1$ for any fixed $r > 1$, that is, fixation is essentially guaranteed when $N$ is large enough. At the same time, the timescale $T(G_N)$ grows quickly in $N$, sometimes even exponentially, thus strong amplification comes at the cost of a substantial slowdown of the evolutionary process. All these structures are either directed or weighted graphs.

Recent research has shown the existence of strong amplifiers that are undirected and unweighted[27,29]. The absence of both weights and directions on the edges is known to lead to timescale that is polynomial in $N$[45,46]. However, this polynomial timescale still remains considerably slower than the logarithmic timescale of the well-mixed population.

**Two theorems.** All existing results on the trade-offs between fixation probability and the timescale highlight two fundamental questions about the evolutionary process. (1) Are there strong amplifiers whose timescale is as short as that of well-mixed populations? And, if not, (2) What is the shortest possible timescale at which strong amplification is possible?

Here we answer the two questions by establishing a tight relationship between the fixation probability and the timescale. First, we show that there exist no strong amplifiers with timescale (asymptotically) comparable to that of complete graphs.

**Theorem 1**
For any strong amplifier $\{G_N\}_{N=1}^{\infty}$, there exists an increasing unbounded function $\beta(N)$ such that

$$T(G_N) \geq \log N \cdot \beta(N). \qquad (3)$$

Hence, as $N \to \infty$, the timescale $T(G_N)$ grows faster than $\log N$, and the graphs $G_N$ incur a slowdown to the evolutionary process compared to the well-mixed population. This answers question (1) in negative and implies that any strong amplifier has to suffer an asymptotically non-negligible slowdown. Later we give a precise expression for $\beta(N)$.

Second, we show that, perhaps surprisingly, the above negative result is tight: For any arbitrarily slowly growing, but unbounded function $\beta(N)$ we construct strong amplifiers that suffer only a slowdown of order $\beta(N)$.

**a  Selection Reactor**
$\mathrm{SR}_N(n, p_{\mathrm{in}}, p_{\mathrm{out}})$

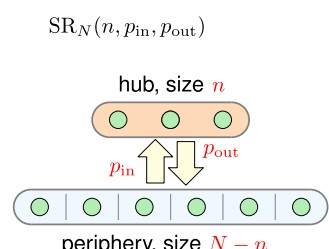

**b  Graph representation**
$\mathrm{SR}_N^{\mathsf{g}}(n, \mathsf{w}_h, \mathsf{w}_\ell)$

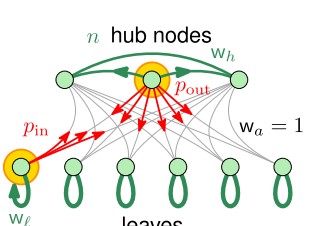

**c  Specific family**
$\mathrm{SR}_N^{\alpha}$

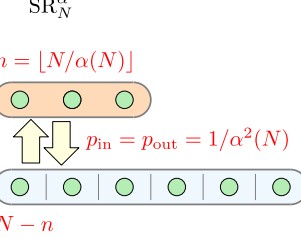

**Fig. 3 Selection reactors. a** A selection reactor $\mathrm{SR}_N(n, p_{\mathrm{in}}, p_{\mathrm{out}})$ consists of two compartments: A central well-mixed 'hub' (orange) containing $n$ nodes and the remaining $N - n$ isolated 'leaves' in the periphery. An offspring of a reproducing individual migrates to the other compartment with probability $p_{\mathrm{in}}$, resp. $p_{\mathrm{out}}$. **b** As a weighted graph, selection reactor $\mathrm{SR}_N^{\mathsf{g}}(n, \mathsf{w}_h, \mathsf{w}_\ell)$ is parametrized by the hub size $n$, the weight $\mathsf{w}_h$ of all edges within the hub, and the weight $\mathsf{w}_\ell$ of all self-loops on the leaves. The migration rates $p_{\mathrm{in}}$ and $p_{\mathrm{out}}$ then satisfy $p_{\mathrm{in}} = n/(n + \mathsf{w}_\ell)$ and $p_{\mathrm{out}} = (N - n)/((N - n) + (n - 1)\mathsf{w}_h)$. **c** Given a slowly growing unbounded function $\alpha(N)$, a selection reactor $\mathrm{SR}_N^{\alpha}$ has $n = \lfloor N/\alpha(N) \rfloor$ nodes in the hub and the migration rates satisfy $p_{\mathrm{in}} = p_{\mathrm{out}} = 1/\alpha^2(N)$ (here $N = 9$, $\alpha(9) = 3$, and $\lfloor \rfloor$ denotes the floor function).

## Theorem 2

For any increasing unbounded function $\beta(N)$ there exists a strong amplifier $\{G_N\}_{N=1}^{\infty}$ such that

$$\mathrm{T}(G_N) \le \log N \cdot \beta(N). \tag{4}$$

This answers question (2) on a positive note and presents strong amplifiers that do not incur an exponential slowdown as compared to the complete graph $K_N$. For instance, by setting $\beta(N) = \log\log N$ we obtain strong amplifiers whose timescale $\log N \cdot \log\log N$ is only marginally longer than the timescale $\log N$ of the well-mixed populations. Together, the two theorems establish a tight dichotomy: All strong amplifiers are asymptotically slower than the complete graphs but any slowdown is sufficient to let strong amplification arise.

The key insight behind Theorem 1 is that the operating principle of any amplifier is to contain a relatively large set $S$ of nodes that are sufficiently isolated so that they are replaced infrequently. Broadly speaking, such nodes are necessary in order to protect the initial mutant from going extinct before it spreads to a sizeable portion of the graph. However, on their way to fixation, the mutants have to spread to all nodes of the graph and the isolated nodes of $S$ now pose a challenge, as the structure protects those nodes from the mutant spread. In particular, we obtain that the slowdown $\beta(N)$ is inversely proportional to the extinction probability $\mathrm{ep}(G_N, r)$, and thus $\beta(N)$ is unbounded for strong amplifiers where $\mathrm{ep}(G_N, r) \to 0$.

We obtain Theorem 2 by introducing a new population structure called the selection reactor, and proving that selection reactors are fast and strong amplifiers.

**Selection reactors.** Selection reactors are population structures motivated by a certain two-chamber system. The population is split into two compartments: a central well-mixed 'hub' and the remaining periphery consisting of isolated 'leaf nodes'. The offspring of a reproducing individual occasionally migrates from one compartment to the other, but the individuals directly compete only when within the hub. A 'selection reactor' $\mathrm{SR}_N(n, p_{\mathrm{in}}, p_{\mathrm{out}})$ is thus fully described by setting four parameters (see Fig. 3a): The total population size $N$, the number $n$ of nodes in the hub, the probability $p_{\mathrm{in}}$ that an offspring of a leaf node migrates to the hub, and the probability $p_{\mathrm{out}}$ that an offspring of a hub node migrates to the periphery.

Mathematically, the structure of a selection reactor can be represented as a weighted graph (see Fig. 3b): Every two of the $n$ hub-nodes are connected by an edge with weight $\mathsf{w}_h$, each of the $N - n$ leafs has a self-loop with weight $\mathsf{w}_\ell$, and every two nodes from different compartments are connected by an edge with

weight $\mathsf{w}_a = 1$ (see Fig. 3b). Setting the weights such that $p_{\mathrm{in}} = n/(n + \mathsf{w}_\ell)$ and $p_{\mathrm{out}} = (N - n)/((N - n) + (n - 1)\mathsf{w}_h)$ we obtain a selection reactor with migration rates $p_{\mathrm{in}}$, $p_{\mathrm{out}}$ as above.

To state our results, it suffices to focus solely on a subfamily $\{\mathrm{SR}^\alpha\}_{N=1}^{\infty}$ of selection reactors parametrized by a function $\alpha : \mathbb{N} \to \mathbb{R}$ which is unbounded ($\alpha(N) \to \infty$ as $N \to \infty$) and slowly growing ($1 < \alpha(N) \ll \sqrt{N}$), see Fig. 3c. Specifically, for any fixed population size $N$, the selection reactor $\mathrm{SR}_N^{\alpha}$ consists of a hub with $n = \lfloor N/\alpha(N) \rfloor$ nodes and the weights of the edges within the hub and of the self-loops are such that whenever a node is selected for reproduction, it replaces a node in the opposite component with probability $p_{\mathrm{in}} = p_{\mathrm{out}} = 1/\alpha^2(N)$. The selection reactor attains its amplification and timescale properties by tuning the function $\alpha(N)$. In particular, we prove that the fixation probability and the timescale satisfy $\mathrm{fp}(\mathrm{SR}_N^{\alpha}, r) \to 1$ and $\mathrm{T}(\mathrm{SR}_N^{\alpha}) \le \log N \cdot \alpha^5(N)$. See the "Methods" section for a high-level overview of the arguments and the Supplementary Note 3 for fully rigorous mathematical proofs.

## Discussion

The fixation probability and the fixation time of mutants are two fundamental quantities of evolutionary dynamics that are affected by a population structure. Traditionally amplifiers of natural selection, which increase the fixation probability of advantageous mutants, also increase the fixation time. Strong amplifiers, which guarantee the fixation of advantageous mutants with high probability, tend to come at the cost of a dramatic increase in fixation time. But the relationship between fixation probability and time has been poorly understood. Here we have shown that a marginal slowdown is necessary for strong amplification, but more than a marginal slowdown is not needed. Thereby we provide a tight resolution for the trade-off between probability and time.

A key component of this resolution is a new class of fast and strong amplifiers, which we call selection reactors. The selection reactor is a population structure with four parameters defining the sizes of its two components (hub and periphery) and the migration rates between them. As we show in the Supplementary Note 3, selection reactors achieve fast and strong amplification for a wide range of those parameters, which makes them a robust structure. Our proof technique might be also applicable to Sparse Incubators[29], which are—to our knowledge—the only other structure that could potentially achieve strong amplification within a comparably short timescale.

So far we have considered the setting where the initial mutant appears at a node selected uniformly a random. This scheme describes situations in which new mutations occur spontaneously at each site with the same rate. Alternatively, for mutations that

occur during reproduction, it is more natural to place the initial mutant at a node selected with probability proportional to the rate at which the node is being replaced by its neighbors. This placing is called temperature initialization[20]. In Supplementary Note 3, we show that both Theorem 1 and Theorem 2 also hold for temperature initialization.

Besides the mathematical appeal of the tight relationship between Theorem 1 and Theorem 2, our positive result on selection reactors has other important implications. First, due to their simplicity, selection reactors could conceivably be constructed in a controlled environment, possibly using microfluidics technology[49,50], or they could occur in natural settings. For example, we speculate that aspects of selection reactor could possibly be at play in germinal centers[51], which are structures involved in the production and affinity maturation of antibodies: Broadly speaking, a germinal center consists of a dark zone and a light zone. Cells occasionally migrate between the two zones but they directly compete with each other only when within the light zone. The light zone thus resembles the hub whereas the dark zone corresponds to the periphery.

Second, note that the fixation probability of a single mutant against a uniform background population of residents characterizes the overall rate of evolution only when each new mutational lineage resolves before the next mutation appears and thus no clonal interference occurs. For structures that operate on a long timescale, the fixation probability is a crucial quantity only when the mutation rate is very low, a restriction that has been received with skepticism[33,36]. In contrast, since Selection Reactors operate on short timescales (relative to other population structures), they avoid clonal interference even with relatively higher mutation rates, and thus their amplifying properties remain relevant for a broader range of parameters. This property lifts the concept of a strong amplifier to a robust phenomenon.

## Methods

Here we informally sketch the intuition behind the proofs of our results. The fully rigorous mathematical proofs can be found in the Supplementary Note 3.

**The slowdown of strong amplifiers**. The key concept behind Theorem 1 is that of temperature. Given two nodes $v$ and $u$, we first denote by $p_{v \to u} = \mathsf{w}_{vu} / \left( \sum_w \mathsf{w}_{vw} \right)$ the probability that when $v$ reproduces, its offspring migrates to $u$ (here $\mathsf{w}_{vu}$ is the weight of the connection between nodes $v$ and $u$). The 'temperature' of a node $u$ is then defined as $\mathcal{T}(u) = \sum_{v \neq u} p_{v \to u}$, that is, it is the (expected) number of times a node $u$ would be replaced by one of its neighbors (per generation) in the neutral setting $r = 1$. When $r > 1$, the replacement rate of a node $u$ slightly changes and it varies depending on which nodes are occupied by mutants, but it is always relatively close to $\mathcal{T}(u)$.

The key step towards Theorem 1 is to show that any strong amplifier contains many nodes, each with a low temperature ("cold" nodes): We show that if ep $(G_N, r) \to 0$, then $G_N$ contains a set $S$ of nodes such that $|S| \propto N$ and such that each node $u \in S$ satisfies

$$\mathcal{T}(u) \propto \mathrm{ep}(G_N, r). \tag{5}$$

The intuition here is that otherwise a substantial portion of newly occurring mutants would be likely to be replaced by one of their neighbors before reproducing even once.

Next we examine the impact of $S$ on the timescale. Initially, at most one node of $S$ is occupied by a mutant. On their way to fixation, mutants have to spread to the whole of $S$. This results in a coupon-collector-like process[52], where it takes increasingly more attempts to spread to the next node in $S$. Since $S$ has size proportional to $N$, mutants will have to make at least of the order of $N \cdot \log N$ attempts to spread into $S$. The low temperature $\mathcal{T}(u)$ of each node $u \in S$ implies that mutants attempt to spread to $u$ only once every $1 / \mathcal{T}(u)$ steps. Overall, this process requires $N \cdot \log N \cdot 1 / \mathcal{T}(u)$ steps or roughly $\log N \cdot \beta(N)$ generations, where

$$\beta(N) \propto \frac{1}{\mathcal{T}(u)} \propto \frac{1}{\mathrm{ep}(G_N, r)} \tag{6}$$

is unbounded since $\mathrm{ep}(G_N, r) \to 0$ as $N \to \infty$.

**Fixation probability analysis**. Given a selection reactor $\mathrm{SR}_N^\alpha$, recall that we denote by $n = \lfloor N/\alpha(N) \rfloor$ the size of the hub and denote by $\ell = N - n$ the number of its leaves. When a leaf node replaces a hub node, we say it 'fires in', likewise a hub node can 'fire out' to replace a leaf. The probabilities $p_{\mathrm{in}}$ (resp. $p_{\mathrm{out}}$) that a reproducing node fires in (resp. out) satisfy $p_{\mathrm{in}} = p_{\mathrm{out}} = 1/\alpha^2(N)$.

First, observe that as $\alpha(N)$ is unbounded, we have $n \ll \ell$, and thus the initial mutant spawns in one of the leaves with high probability $\ell/N$. Second, this initial mutant can disappear only due to a resident firing out from the hub. In any one step, this event has probability roughly $p^- \propto \frac{n}{N} \cdot p_{\mathrm{out}} \cdot \frac{1}{\ell} \propto \frac{1}{\alpha^3(N) \cdot N}$, whereas the initial mutant fires in the hub with probability roughly $p^+ \propto \frac{1}{N} \cdot p_{\mathrm{in}} \propto \frac{1}{\alpha^2(N) \cdot N}$. Thus the initial mutant has enough time to place offspring in the hub roughly $p^+/(p^+ + p^-) \propto \alpha(N)$ times. Third, we argue that mutants spread to a half of the hub: We show that with fewer than $n/2$ mutants in the hub, in each step there is a bias towards gaining a mutant in the hub rather than losing one. This implies that each of the roughly $\alpha(N)$ offsprings of the first mutant has a constant chance of spreading to a half of the hub. We then show that, with high probability, at least one of the offspring does so. Fourth, we show that from this point on, mutant fixation is almost guaranteed, as any attempt from the residents to regain the hub has to convert back $n/2$ mutants, which is exponentially unlikely due to its well-mixed structure. Thus $\mathrm{SR}_N^\alpha$ achieves strong amplification.

**Timescale analysis**. To understand the timescale of selection reactors, we decompose the evolutionary process into two components.

(1) The *active process*, which tracks the active steps of the Moran process, where the population changes state (i.e., a mutant replaces a resident or vice-versa).
(2) The *waiting process*, which tracks the steps of the process where the population state does not change.

We then define a potential function $\psi$ that assigns a number to each state and attains its maximum value upon fixation. In expectation, $\psi$ increases by a constant in each active step but it can fluctuate. Nevertheless, we show that $\psi$ will attain any specific value $k$ at most $\alpha^2(N)$ times, in expectation. Next, we denote by $W_k$ the maximum number of expected waiting steps that the process makes when in a state with potential equal to $k$. Although $W_k$ depends on $k$, we show that asymptotically

$$\sum_k W_k \leq N \cdot \log N \cdot \alpha^3(N). \tag{7}$$

Since every time the process is in a state with potential $k$, it will wait for at most $W_k$ steps in expectation, and any such state is visited at most $\alpha^2(N)$ times in expectation, we obtain that total expected number of steps for the selection reactor is at most

$$\sum_k \alpha^2(N) \cdot W_k \leq N \cdot \log N \cdot \alpha^5(N). \tag{8}$$

Dividing this by $N$ gives the timescale of the process

$$\mathrm{T}(\mathrm{SR}_N^\alpha) \leq \log N \cdot \alpha^5(N) = \log N \cdot \beta(N), \tag{9}$$

by choosing $\alpha(N) = \sqrt[5]{\beta(N)}$.

**Organization of the supplementary information**. The organization of the supplementary information is as follows: In Supplementary Note 1, we introduce the necessary terminology and notation, together with the mathematical tools we are using in the proofs. In Supplementary Note 2, we formally state our mathematical results and provide a high-level overview of the ideas behind the proofs. In Supplementary Note 3, we then provide those formal proofs.

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

## Acknowledgements

K.C. acknowledges support from ERC Start grant no. (279307: Graph Games), ERC Consolidator grant no. (863818: ForM-SMart), Austrian Science Fund (FWF) grant no. P23499-N23 and S11407-N23 (RiSE). M.A.N. acknowledges support from Office of Naval Research grant N00014-16-1-2914 and from the John Templeton Foundation.

## Author contributions

J.T., A.P., K.C., and M.A.N. designed the research, performed the analysis, and wrote the manuscript.

## Competing interests

The authors declare no competing interests.
