## [Peer Review File · Nature Communications]

Reviewers' Comments:

Reviewer #1:

Remarks to the Author:

In "Fast and strong amplifiers of natural selection" authors study selection and random drift, which determine the probability that novel mutations fixate in a population. Previous research has identified population structures that increase the fixation probability of beneficial mutants compared to well-mixed populations, but at a cost of a slow fixation process. Indeed, not only was the velocity slow, but also the precise relationship between fixation probability and time has remained elusive. In their seminal contributions, authors now for the first time provide precise results that characterize the slowdown effect of strong amplification.

With the new research, the authors are able to prove that in fact all strong amplifiers delay the fixation event at least to some extent. However, they were able to construct strong amplifiers that delay the fixation event only marginally compared to the well-mixed populations.

How and why fixation emerges in natural selection is an intensely investigated subject with obvious practical ramifications. Methods of theoretical biology, applied mathematics, statistical physics, and network science have been used successfully and with much effect in recent years to shed light on the problem from many different perspectives, and also to outline many different ways on how fixation could be better understood, and more generally, how it could be promoted under various constraints in terms of population structure and mutant abundance. This study thus certainly addresses and significantly advances an important subject, and it also delivers results that will surely be of interest to the readership of Nature Communications.

I have enjoyed reading this manuscript. I find it comprehensive and clearly written, and introducing new fundamental results that will surely also inspire future research along similar lines, in particular to further promote better theoretical foundations between fixation probability and time. For the time being, the final message, which is that "strong amplification always comes at a cost of a slowdown, but more than a marginal slowdown is not needed" is certainly inspiring and worth recognition that publication in Nature Communications confers.

A revision should kindly address the following comments.

In terms of related research, there are two recent papers by Yang et al. titled Strategically positioning cooperators can facilitate the contagion of cooperation, *Sci. Rep.* 11, 1127 (2021) and Identification of influential invaders in evolutionary populations, *Sci. Rep.* 9, 7305 (2019), where fixation has been studied in relation to placement in structured populations.

It would also be welcome to spell out limitations in terms of the size of the population for which we can still hope to obtain useful results with reasonable resources in terms of computational time. In particular, can the authors perhaps present a scaling relation for the time needed in dependence on population size?

Apart from this, I am happy to congratulate the authors to an excellent contribution.

Reviewer #2:

Remarks to the Author:

The authors study a birth-death dynamics on finite structured populations. They primarily deal with strong amplifiers which are structures that enhance the natural selection of advantageous mutants. They prove that all strong amplifiers must delay the mutant fixation (compared to well mixed population). They also construct strong amplifiers that delay the fixation only marginally.

This is a very well written manuscript. The math results are very elegant and clean.

The manuscript can be accepted as is.

Reviewer #3:

Remarks to the Author:

Please see my comments in the attached pdf.

Review of Tkadlec *et. al.*, *Fast and strong amplifiers of natural selection*

The authors show that the fixation time for the Moran model in an well-mixed population of fixed size is an asymptotic lower bound on the time to fixation for the Moran model on strong amplifier graphs (weighted graphs such that the probability of fixation of any strongly selected mutant approaches 1 as the population size N tends to infinity, for any value of individual fitness exceeding 1). They moreover show that this bound is tight by constructing a family of graphs, *Selection Reactors*, which can achieve fixation times bounded above by any function asymptotically larger than $\ln N$, the order of the fixation time in the well-mixed Moran model. The results are pleasing, and the arguments are generally clear and well developed. In my reading I came across what seem to be to be minor concerns with the proofs, but these should be straightforward to resolve. I quite appreciated the inclusion of intuitive sketches of the proofs in both main text and in the supplement, as well as the expositional material in the supplement (some comments on this below), although I personally found the sketch in the SI to be clearer than that in the main text and found myself wondering if two separate sketches were necessary.

On a more general level, my concern with this manuscript (and indeed the entire subfield of evolutionary graph theory) is that it seems to be slipping into the realm of pure mathematics, constructing ever more elaborate structures with some property or other without a sufficiently compelling argument for the relevance to the biological questions that inspired them. Pure mathematics is an eminently worth endeavour, but given that this was submitted to *Nature Communications*, and not, say *Random Structures & Algorithms*, begs the question of broader relevance: how generic is the property of being a fast & strong amplifier? Do we expect a set of positive density in the space of graphs to have the property? Is there a plausible mechanism by which evolution or self-organization might organize a population into a structure with these properties? Germinal centres are given as an example of a population with a plausible hub/leaf structure, but is there any evidence that self-replacement occurs, but exclusively in the leaves in that or any other biological example? Can an example be constructed without the necessity of such specific self-replacement, *e.g.* by replacing leaves with islands in a mainland-island meta-community model? How large does N have to be to get fast & strong amplification? The argument hinges on the fact that the mutant appears in the leaves *w.h.p.*, but, by choosing a sufficiently slowly growing function $\alpha(N)$, one can ensure that the N required is astronomical (if I take $\alpha(N) = \log_{10}(\log_{10}(N))$, then I would need a googol of individuals in order to achieve a probability of 90% that the mutant appears in the leaves, several

orders of magnitude greater than the number of elementary particles in the observable universe). None of these diminishes the mathematical interest of the result, but the authors appear to be claiming biological relevance as well, and I don't feel the arguments for the latter are sufficiently well developed. For example, on page 6, it is asserted that "In contrast, since Selection Reactors operate on short timescales, their amplifying properties are relevant also for higher mutation rates. This property lifts the concept of a strong amplifier from arguably a mathematical curiosity to a robust phenomenon." In addition to the questions raised above concerning the actual speed and amplification achieved in populations of plausible size and structure, I would argue that making claims about higher mutation rates demands serious consideration of the question of clonal interference, which I suspect would complicate things considerably. In brief, for this journal and this audience, I would like to see more evidence that this is more than a "mathematical curiosity", if an appealing one at that.

Comments on the SI and Proofs (ordered chronologically, and not by importance)

- In defining the temperature initialization, it might be worth mentioning that the quantities $\mathcal{T}(v) + \mathcal{L}(v)$ do indeed sum to N .
- On page 3, it seems to me more appropriate to cite the original authors, *i.e.* Moran, or standard texts, *e.g.* Ewens, for classical results on the fixation probability and time for the Moran model, rather than self-cite,
- "arbitrarily minute fitness advantage $r > 1$ " should be qualified by the fact that r is assumed independent of N and that the results remain in the domain of strong selection.
- For present purposes, it seems sufficient, and clearer for a broader audience, to limit the definition of (sub)martingales to the Markov case: $\mathbb{E}[M_t | M_s] = M_s$. It might also be appropriate to define a (Markov) stopping time here, as the concept is used relatively frequently.
- Just to repeat, this expository section is a welcome addition, as are the following proof sketches.
- I don't believe that this statement is correct: "the modified process M' always terminates with the mutants fixating and its expected fixation time is given by $FT'(G, r, u) = \frac{1}{fp(G, r, u)} FT(G, r, u)$ ". I'm assuming the argument is that there are a geometrically distributed number of independent "attempts" for the mutant to fix starting from vertex u , with success parameter $fp(G, r, u)$; if that is the reasoning, it seems that one needs to take into account that all but the last, successful attempt end in fixation of the resident, and that, as the authors note previously, the fixation time conditioned on the fixation of the mutant is asymptotically longer than the fixation time conditioned on fixation of the resident. It seems simpler, and sufficient for present purposes to observe that

$$\begin{aligned} \mathbb{E}[\text{fixation time}] &= fp(G, r, u) \mathbb{E}[\text{fixation time} | \text{mutant fixes}] \\ &\quad + (1 - fp(G, r, u)) \mathbb{E}[\text{fixation time} | \text{resident fixes}] \\ &\geq fp(G, r, u) \mathbb{E}[\text{fixation time} | \text{mutant fixes}]. \end{aligned}$$

The argument deriving a lower bound for $FT'(G, r, u)$ applies equally well to bounding the conditional fixation time.

- Perhaps a matter of personal preference, but it seems to me clearer to speak of “absorption time” and limit “fixation time” to refer to fixation of the mutant, to be consistent with the term “fixation probability”.
- The phrase “nodes with small extinction probability” appears before is defined Lemma 2 to refer to the extinction probability conditional on the node in which the mutant initially appears. Please give a clear definition prior.
- Starting with Lemma 2, notation seems to be used increasingly inconsistently, *e.g.* the process X_t is introduced without definition as the configuration at time t , whereas previously \mathcal{S}_t had been used, also, the fitness given configuration S is alternately written as F , F_S , and $F(S)$.
- At the start of §4.2: “Let $S_{i,j}$ be a configuration. . .”
- In the statement of Lemma 4: “Let $S_{i',j'}$ be the first configuration with $j' \neq j$.” At the risk of being pedantic, this should be specified as the first configuration with $j' \neq j$ attained by the Markov chain starting from the initial state $S_{i,j}$.
- At the top of page 12, the definition of p_{in} and p_{out} appear to have changed from both being $\frac{1}{\alpha^2(N)}$ to being $\frac{1}{\alpha^2(m)}$ and $\frac{1}{\alpha^2(\ell)}$, respectively. m is undefined. Also, why the change from writing N to writing $\ell + n$ in what follows?
- Lemma 6 could be perhaps better expressed explicitly as proof by contradiction.
- In Lemma 8, after having gone to the effort to define submartingales, why is $\phi(\mathcal{S}_t)$ never identified as such?
- $\gamma = \frac{hr+1}{hr+r^2} = 1 - \frac{r^2-1}{hr+r^2} \rightarrow 1$, not $1 - \frac{r^2-1}{hr}$; the latter is asymptotically equivalent to γ , but is not a valid limit.
- In the same lemma, the quantity $\phi(\mathcal{S}_\infty)$ (here written as ϕ^∞) needs to be taken in expected value: $\mathbb{E}[\phi(\mathcal{S}_\infty)]$ (and, if one wants to give a “fully rigorous proof” as claimed, should be better identified as the $\mathbb{E}[\phi(\mathcal{S}_T)]$, where T is the absorption time, an example of a stopping time, and then it should be verified that \mathcal{S}_t and T satisfy one of the criteria for Doob’s optional stopping theorem).
- The statement in the main text $h < \sqrt{N}$ needs to be made consistent with the actual condition of Theorem 2, $h = o(\sqrt{N})$.
- A matter of personal preference, but Theorem 2 is immediate from the four lemmas.
- In the statement of Lemma 9, the expectation $\mathbb{E}[\psi(S')]$ needs to be made conditional on the initial condition S . My own feeling is that many of these statements would be more precisely (better) expressed consistently in terms of a stochastic process \mathcal{S}_t , its initial condition, and stopping times that indicate the first time a given (class of) configuration(s) has been achieved.

- More importantly, to prove the submartingale property here, I don't believe it is sufficient to consider the various events along a given edge pairwise: as currently phrased, one considers pairs of events $p_{u \rightarrow v}$ and $p_{v \rightarrow u}$ and then using the ratio *e.g.* $\frac{p_{u \rightarrow v}}{p_{u \rightarrow v} + p_{v \rightarrow u}}$ for the probability that $u \rightarrow v$ is the event that happens first. I think that one needs to consider all possible events that could change the configuration, and asking which of these occurs first, and not simply pairs of events along a given edge.
- In Lemma 11, the function $\xi(\mathcal{S}_0, \dots, \mathcal{S}_t)$ doesn't add much other than confusion. $\psi(\mathcal{S}_t) - ct$ is already sufficient to define the submartingale, and due to the Markov property, conditioning on the whole sigma-algebra $\mathcal{F}_t = \sigma(\{\mathcal{S}_0, \dots, \mathcal{S}_t\})$ is equivalent to conditioning on \mathcal{S}_t , so the additional history is irrelevant.
- Lemma 12: "slices" = subsets.

Point-by-point response

Reviewer #1

Ans: We thank the reviewer for a positive review. We answer the two comments below.

In "Fast and strong amplifiers of natural selection" authors study selection and random drift, which determine the probability that novel mutations fixate in a population. Previous research has identified population structures that increase the fixation probability of beneficial mutants compared to well-mixed populations, but at a cost of a slow fixation process. Indeed, not only was the velocity slow, but also the precise relationship between fixation probability and time has remained elusive. In their seminal contributions, authors now for the first time provide precise results that characterize the slowdown effect of strong amplification.

With the new research, the authors are able to prove that in fact all strong amplifiers delay the fixation event at least to some extent. However, they were able to construct strong amplifiers that delay the fixation event only marginally compared to the well-mixed populations.

How and why fixation emerges in natural selection is an intensely investigated subject with obvious practical ramifications. Methods of theoretical biology, applied mathematics, statistical physics, and network science have been used successfully and with much effect in recent years to shed light on the problem from many different perspectives, and also to outline many different ways on how fixation could be better understood, and more generally, how it could be promoted under various constraints in terms of population structure and mutant abundance. This study thus certainly addresses and significantly advances an important subject, and it also delivers results that will surely be of interest to the readership of Nature Communications.

I have enjoyed reading this manuscript. I find it comprehensive and clearly written, and introducing new fundamental results that will surely also inspire future research along similar lines, in particular to further promote better theoretical foundations between fixation probability and time. For the time being, the final message, which is that "strong amplification always comes at a cost of a slowdown, but more than a marginal slowdown is not needed" is certainly inspiring and worth recognition that publication in Nature Communications confers.

A revision should kindly address the following comments.

In terms of related research, there are two recent papers by Yang et al. titled Strategically positioning cooperators can facilitate the contagion of cooperation, *Sci. Rep.* 11, 1127 (2021) and Identification of influential invaders in evolutionary populations, *Sci. Rep.* 9, 7305 (2019), where fixation has been studied in relation to placement in structured populations.

Ans: Thank you for pointing out these relevant references, we have added them.

It would also be welcome to spell out limitations in terms of the size of the population for which we can still hope to obtain useful results with reasonable resources in terms of computational time. In particular, can the authors perhaps present a scaling relation for the time needed in dependence on population size?

Ans: We can show that the necessary slowdown $\beta(N)$ is (at least) inversely proportional to the extinction probability, see the first paragraph of section Methods. For a strong amplifier, the extinction probability must tend to 0, but it does so at different rates for different structures. For instance, the family M_N of Megastars introduced in [29] satisfies $\epsilon_p(M_N, r) = \Theta(N^{-1/2})$ and thus their slowdown is at least $\Omega(N^{1/2})$.

Apart from this, I am happy to congratulate the authors to an excellent contribution.

Ans: Thank you for nice words.

Reviewer #2

Ans: We thank the reviewer for a positive review.

The authors study a birth-death dynamics on finite structured populations. They primarily deal with strong amplifiers which are structures that enhance the natural selection of advantageous mutants. They prove that all strong amplifiers must delay the mutant fixation (compared to well mixed population). They also construct strong amplifiers that delay the fixation only marginally.

This is a very well written manuscript. The math results are very elegant and clean.

The manuscript can be accepted as is.

Reviewer 3

Ans: We thank the reviewer for a thorough review and many good suggestions, in particular regarding the SI and the proofs there. We answer the comments below.

The authors show that the fixation time for the Moran model in an well-mixed population of fixed size is an asymptotic lower bound on the time to fixation for the Moran model on strong amplifier graphs (weighted graphs such that the probability of fixation of any strongly selected mutant approaches 1 as the population size N tends to infinity, for any value of individual fitness exceeding 1). They moreover show that this bound is tight by constructing a family of graphs, Selection Reactors, which can achieve fixation times bounded above by any function asymptotically larger than $\ln N$, the order of the fixation time in the well-mixed Moran model. The results are pleasing, and the arguments are generally clear and well developed. In my reading I came across what seem to be to be minor concerns with the proofs, but these should be straightforward to resolve. I quite appreciated the inclusion of intuitive sketches of the proofs in both main text and in the supplement, as well as the expositional material in the supplement (some

comments on this below), although I personally found the sketch in the SI to be clearer than that in the main text and found myself wondering if two separate sketches were necessary.

Ans: Thank you for nice words.

On a more general level, my concern with this manuscript (and indeed the entire subfield of evolutionary graph theory) is that it seems to be sipping into the realm of pure mathematics, constructing ever more elaborate structures with some property or other without a sufficiently compelling argument for the relevance to the biological questions that inspired them. Pure mathematics is an eminently worth endeavour, but given that this was submitted to Nature Communications, and not, say Random Structures & Algorithms, begs the question of broader relevance: how generic is the property of being a fast & strong amplifier? Do we expect a set of positive density in the space of graphs to have the property? Is there a plausible mechanism by which evolution or self-organization might organize a population into a structure with these properties? Germinal centres are given as an example of a population with a plausible hub/leaf structure, but is there any evidence that self-replacement occurs, but exclusively in the leaves in that or any other biological example? Can an example be constructed without the necessity of such specific self-replacement, e.g. by replacing leaves with islands in a mainland-island meta-community model? How large does N have to be to get fast & strong amplification? The argument hinges on the fact that the mutant appears in the leaves w.h.p, but, by choosing a sufficiently slowly growing function $\alpha(N)$, one can ensure that the N required is astronomical (if I take $\alpha(N) = \log_{10}(\log_{10}(N))$, then I would need a googol of individuals in order to achieve a probability of 90% that the mutant appears in the leaves, several 1 orders of magnitude greater than the number of elementary particles in the observable universe). None of these diminishes the mathematical interest of the result, but the authors appear to be claiming biological relevance as well, and I don't feel the arguments for the latter are sufficiently well developed. For example, on page 6, it is asserted that "In contrast, since Selection Reactors operate on short timescales, their amplifying properties are relevant also for higher mutation rates. This property lifts the concept of a strong amplifier from arguably a mathematical curiosity to a robust phenomenon." In addition to the questions raised above concerning the actual speed and amplification achieved in populations of plausible size and structure, I would argue that making claims about higher mutation rates demands serious consideration of the question of clonal interference, which I suspect would complicate things considerably. In brief, for this journal and this audience, I would like to see more evidence that this is more than a "mathematical curiosity", if an appealing one at that.

Ans: Those are great questions. We view the present manuscript as a proof of concept. We admit that actual biological populations are more complicated than the model. Therefore, we toned down the text in several places. At the same time, we believe that our results provide an excellent starting point for future theoretical and empirical studies and that Selection Reactors in particular are promising structures. In more detail:

- elaborate structure/how generic: We think that Selection Reactors are in fact less elaborate and more robust than most other structures that were already studied. For instance, in a Star or a Superstar graph, removing a single node (the center) completely breaks the amplification properties. In contrast, Selection Reactors contain no such distinguished nodes, just two large patches. Moreover, even though in the main text we focus on a single specific family of Selection Reactors, our proofs in the SI go through for a more generic structure that we call Reactors. This class of structures has a range of weights and the relative sizes of the patches, for which our Theorem B also applies (and conceivably there is an even broader range where our proofs cease to work but Reactors still have the desired properties).

- positive density in the space of graphs: For small population sizes, it is known that most structures provide some amount of amplification (see [36], [48]). We believe that strong amplification is

relatively rare but since Selection Reactors are robust, they form a relatively broad peak (in the imaginary high-dimensional landscape of population structures) and an evolutionary search could be able to locate them.

- germinal centers: We mention germinal centers as an example of a two-patch structure that seems to exhibit some common features with Reactors. We do not know (or claim) that the underlying mechanisms behind how the two structures operate are the same, and we now emphasized this in the text. The good news is that since Reactors are robust, some self-replacement in the hub (or the light zone) is still fine without affecting Theorem B.

- self-loops: We think that fast and strong amplifiers with no self-loops at all do exist: In particular, we think that one might be able to adapt the proof for Reactors to the (appropriately tuned) family of Sparse Incubators (see [29]). Also, self-loops might not be as strange biologically as they might look mathematically -- in a cell division, the parent cell is replaced by the daughter cell.

- how large does N have to be?: It is true that we chose to provide only an asymptotic result (in the limit $N \rightarrow \infty$), as this gives a clear qualitative message. As a consequence, the specific family of Reactors that we talk about in the main text is optimized for illustrating the qualitative claim, not for showing the limiting behavior for small N . We think that exploring tradeoffs such as this one is a good direction for subsequent work.

- clonal interference: The quoted claim is meant to state that thanks to the short fixation time (relative to other population structures) structures, Reactors "withstand" relatively high mutation rates without clonal interference occurring. We don't make any claims about mutation rates that are so high that clonal interference occurs and we now emphasized that. We also agree that studying an analogous evolutionary process under clonal interference is considerably more complicated and as such we think it is outside the scope of this work -- but, again, a great direction for future work.

- mathematical curiosity: We see our paper as more than a mathematical curiosity for the following reasons: The question how natural selection is affected by population structure is of fundamental importance in biology. Reactors are simple and robust structures. They are also much faster than previous amplifiers of selection. We hope that our results will inspire biologists to look differently at real population structures and follow up with computer simulations and lab experiments.

Comments on the SI and Proofs (ordered chronologically, and not by importance)

Ans: We thank the reviewer for many good comments on typos or insufficient explanations in the SI (we fixed all of them) and suggestions (we incorporated most of them).

• In defining the temperature initialization, it might be worth mentioning that the quantities $T(v)+L(v)$ do indeed sum to N .

Ans: Done.

• On page 3, it seems to me more appropriate to cite the original authors, i.e. Moran, or standard texts, e.g. Ewens, for classical results on the fixation probability and time for the Moran model, rather than self-cite,

Ans: Done (also in the main text; we keep one self-citation for the explicit computation of the lower order terms in fixation time).

• "arbitrarily minute fitness advantage $r > 1$ " should be qualified by the fact that r is assumed independent of N and that the results remain in the domain of strong selection.

Ans: Done.

- For present purposes, it seems sufficient, and clearer for a broader audience, to limit the definition of (sub)martingales to the Markov case: $E[M_t | M_s] = M_s$. It might also be appropriate to define a (Markov) stopping time here, as the concept is used relatively frequently.

Ans: As suggested, we limit the definition to the Markov case.

- Just to repeat, this expository section is a welcome addition, as are the following proof sketches.

Ans: Thank you.

- I don't believe that this statement is correct: "the modified process M_0 always terminates with the mutants fixating and its expected fixation time is given by $FT'(G,r,u) = 1/fp(G,r,u) \cdot FT(G,r,u)$ ". I'm assuming the argument is that there are a geometrically distributed number of independent "attempts" for the mutant to fix starting from vertex u , with success parameter $fp(G,r,u)$; if that is the reasoning, it seems that one needs to take into account that all but the last, successful attempt end in fixation of the resident, and that, as the authors note previously, the fixation time conditioned on the fixation of the mutant is asymptotically longer than the fixation time conditioned on fixation of the resident. It seems simpler, and sufficient for present purposes to observe that $E[\text{fixation time}] = fp(G,r,u)E[\text{fixation time}|\text{mutant fixes}] + (1-fp(G,r,u))E[\text{fixation time}|\text{resident fixes}] \geq fp(G,r,u)E[\text{fixation time}|\text{mutant fixes}]$. The argument deriving a lower bound for $FT'(G,r,u)$ applies equally well to bounding the conditional fixation time.

Ans: Two points: (i) We believe that the statement is correct. Our original argument is (implicitly) by linearity of expectation (there is no need to take into account that only the last attempt is successful). However, we agree that the writeup was too brief and since the argument is kind of subtle, instead we now compute the fixation time of M' explicitly.

(ii) In the suggested workaround, the inequality $E[\text{fixation time}] \geq fp \cdot E[\text{fixation time}|\text{mutant fixes}]$ is true but we don't see an easy way to use it to get a lower bound on fixation time: When conditioning on (ultimate) fixation of the mutant, one can not simply sum up the "waiting times" at different configurations using linearity of expectations (one would have to "weight" those contributions by certain fixation probabilities). As an example, think of a Markov chain with four states [FixRes], [A], [B], [FixMut] and transition probabilities as follows:

Consider a random walk starting at [A] and let C be the expected number of steps until it reaches [FixMut], conditioning on the fact that it does so (before reaching [FixRes]). Summing over all relevant trajectories we get

$$C = \frac{\sum_{k=1}^{\infty} 2k \cdot (9/100)^{k-1} \cdot (1/100)}{\sum_{k=1}^{\infty} (9/100)^{k-1} \cdot (1/100)} = \frac{200/91^2}{1/91} = 200/91 \approx 2$$

which is less than $10+10$. Intuitively, this is because by conditioning on fixation of the mutant, we pushed most of the weight to the “lucky” trajectory that terminates in 2 steps.)

- Perhaps a matter of personal preference, but it seems to me clearer to speak of “absorption time” and limit “fixation time” to refer to fixation of the mutant, to be consistent with the term “fixation probability”.

Ans: We agree that this is a matter of personal preference (we prefer to keep the terminology “(conditional) fixation time”), as it is common in related literature.

- The phrase “nodes with small extinction probability” appears before is defined Lemma 2 to refer to the extinction probability conditional on the node in which the mutant initially appears. Please give a clear definition prior.

Ans: Done.

- Starting with Lemma 2, notation seems to be used increasingly inconsistently, e.g. the process X_t is introduced without definition as the configuration at time t , whereas previously S_t had been used, also, the fitness given configuration S is alternately written as F , F_S , and $F(S)$.

Ans: Done.

- At the start of §4.2: “Let $S_{i,j}$ be a configuration. . .”

Ans: Done.

- In the statement of Lemma 4: “Let $S_{\{i,j\}}$ be the first configuration with $j \neq j$.” At the risk of being pedantic, this should be specified as the first configuration with $j' = j$ attained by the Markov chain starting from the initial state $S_{\{i,j\}}$.

Ans: Done (also in Lemma 6).

- At the top of page 12, the definition of p_{in} and p_{out} appear to have changed from both being $1/\alpha^2(N)$ to being $1/\alpha^2(m)$ and $1/\alpha^2(l)$, respectively. m is undefined. Also, why the change from writing N to writing $l+n$ in what follows?

Ans: Done.

- Lemma 6 could be perhaps better expressed explicitly as proof by contradiction.

Ans: We prefer to keep the proof “direct”.

- In Lemma 8, after having gone to the effort to define submartingales, why is $\phi(S_t)$ never identified as such?

Ans: Done.

• $\gamma = \frac{hr+1}{hr+r} = \frac{1-r}{2-1} = \frac{hr+r}{2} \rightarrow 1$, not $\frac{1-r}{2-1} = \frac{hr}{2}$; the latter is asymptotically equivalent to γ , but is not a valid limit.

Ans: Done.

• In the same lemma, the quantity $\phi(S^\infty)$ (here written as ϕ^∞) needs to be taken in expected value: $E[\phi(S^\infty)]$ (and, if one wants to give a “fully rigorous proof” as claimed, should be better identified as the $E[\phi(S_T)]$, where T is the absorption time, an example of a stopping time, and then it should be verified that S_t and T satisfy one of the criteria for Doob’s optional stopping theorem).

Ans: Done.

• The statement in the main text $h < \sqrt{N}$ needs to be made consistent with the actual condition of Theorem 2, $h = o(\sqrt{N})$.

Ans: Done.

• A matter of personal preference, but Theorem 2 is immediate from the four lemmas.

Ans: We prefer to keep the (short) proof included, to tie the lemmas together.

• In the statement of Lemma 9, the expectation $E[\psi(S')]$ needs to be made conditional on the initial condition S . My own feeling is that many of these statements would be more precisely (better) expressed consistently in terms of a stochastic process S_t , its initial condition, and stopping times that indicate the first time a given (class of) configuration(s) has been achieved.

Ans: As suggested, we replaced $E[\psi(S')]$ with $E[\psi(S') | S]$

• More importantly, to prove the submartingale property here, I don’t believe it is sufficient to consider the various events along a given edge pairwise: as currently phrased, one considers pairs of events $pu \rightarrow v$ and $pv \rightarrow u$ and then using the ratio e.g. $\frac{pu \rightarrow v}{pu \rightarrow v + pv \rightarrow u}$ for the probability that $u \rightarrow v$ is the event that happens first. I think that one needs to consider all possible events that could change the configuration, and asking which of these occurs first, and not simply pairs of events along a given edge.

Ans: We believe the argument is correct. We expanded on it to clarify. (Intuitively, the point is that each active step happens along some active edge, so we can condition on the edge being used and then the “local” bounds along each edge give a “global” bound for the whole process: If $A \geq a$ and $B \geq b$ then $\alpha A + \beta B \geq \alpha a + \beta b$.)

• In Lemma 11, the function $\xi(S_0, \dots, S_t)$ doesn’t add much other than confusion. $\psi(S_t) - ct$ is already sufficient to define the submartingale, and due to the Markov property, conditioning on the whole sigma-algebra $\mathcal{F}_t = \sigma(\{S_0, \dots, S_t\})$ is equivalent to conditioning on S_t , so the additional history is irrelevant.

Ans: Done, thank you.

• Lemma 12: “slices” = subsets.

Ans: Done.

Reviewers' Comments:

Reviewer #1:

Remarks to the Author:

I would like to thank the authors for addressing my minor comments constructively and in detail. I warmly recommend publication in present form.

Reviewer #2:

Remarks to the Author:

The authors addressed all issues appropriately.